# Predicting the Effect of Single Mutations on Protein Stability and Binding with Respect to Types of Mutations

**DOI:** 10.3390/ijms241512073

**Published:** 2023-07-28

**Authors:** Preeti Pandey, Shailesh Kumar Panday, Prawin Rimal, Nicolas Ancona, Emil Alexov

**Affiliations:** 1Department of Physics and Astronomy, Clemson University, Clemson, SC 29634, USA; preetip@g.clemson.edu (P.P.); spanday@clemson.edu (S.K.P.); primal@g.clemson.edu (P.R.); 2Department of Biological Sciences, Clemson University, Clemson, SC 29634, USA; nancona@g.clemson.edu

**Keywords:** mutations, folding free energy change, binding free energy change, single nucleotide variant

## Abstract

The development of methods and algorithms to predict the effect of mutations on protein stability, protein–protein interaction, and protein–DNA/RNA binding is necessitated by the needs of protein engineering and for understanding the molecular mechanism of disease-causing variants. The vast majority of the leading methods require a database of experimentally measured folding and binding free energy changes for training. These databases are collections of experimental data taken from scientific investigations typically aimed at probing the role of particular residues on the above-mentioned thermodynamic characteristics, i.e., the mutations are not introduced at random and do not necessarily represent mutations originating from single nucleotide variants (SNV). Thus, the reported performance of the leading algorithms assessed on these databases or other limited cases may not be applicable for predicting the effect of SNVs seen in the human population. Indeed, we demonstrate that the SNVs and non-SNVs are not equally presented in the corresponding databases, and the distribution of the free energy changes is not the same. It is shown that the Pearson correlation coefficients (PCCs) of folding and binding free energy changes obtained in cases involving SNVs are smaller than for non-SNVs, indicating that caution should be used in applying them to reveal the effect of human SNVs. Furthermore, it is demonstrated that some methods are sensitive to the chemical nature of the mutations, resulting in PCCs that differ by a factor of four across chemically different mutations. All methods are found to underestimate the energy changes by roughly a factor of 2.

## 1. Introduction

Biological macromolecules, such as proteins, DNA, and RNAs, perform their functions by adopting a particular 3D structure and being involved in a set of interactions. For many proteins, excluding intrinsically disordered proteins (IDPs), the correctly folded 3D structure is needed to prevent them from protease degradation and to form the desired catalytic set of residues, binding interface, and other functionally important structural features [1,2]. The assessment of the stability of such a 3D structure is conducted via a thermodynamic quantity called folding free energy, i.e., the difference between folded free and unfolded free energies (ΔG_folding_). Another important process is the binding of biological macromolecules, at which they adopt particular 3D complexes, including those of IDPs, which upon binding form a well-defined 3D structure [3,4]. Similarly, as above, the ability of macromolecules to form a macromolecular complex is assessed via binding free energy (ΔG_binding_), i.e., the difference between the free energy of bound and unbound states. Thus, because of their importance for the biological function of macromolecules, ΔG_folding_ and ΔG_binding_ were extensively investigated experimentally, and many methods for predicting them were developed [5,6,7,8,9,10].

The vast majority of the experimental works were conducted to assess the impact of a given residue on either ΔG_folding_ or ΔG_binding_, involving the substitution of wild-type residues to alanine (alanine scanning) [11,12,13]. This raises the question of the balance between investigator-initiated mutations and mutations seen in nature, i.e., in the human population, which are single nucleotide variants (SNVs). It should be mentioned that mutations and SNVs are both types of genetic variations that can occur in the DNA sequence (this article focuses on missense mutations, i.e., mutations that result in a change in the amino acid sequence of the corresponding protein). However, there is a subtle difference between the two terms, since a mutation is a broader term that refers to any change in the DNA sequence that is different from the wild type or the reference sequence, while SNV is a specific type of mutation that involves the substitution of a single nucleotide (A, T, C, or G) at a specific position in the DNA sequence. Thus, SNVs are a type of mutation, but not all mutations are SNVs. In this article, we will provide an assessment of the distribution of SNVs and non-SNVs and the corresponding free energy changes reported in the most popular databases.

Here we briefly outline some of the popular databases of experimentally measured thermodynamic quantities related to protein stability, protein–DNA interaction, and protein–protein binding often used by researchers for developing and assessing the performance of new methods for predicting the stability of proteins and their interactions with other proteins and/or DNA. ProTherm [14,15] is a database that consists of the experimentally measured ΔG_folding_ of wild-type proteins along with single and multiple mutations. In addition, it also provides information about the experimental conditions, such as pH and temperature. ProNIT and ProNAB are databases of experimentally determined protein-nucleic acid ΔG_binding_ [16,17]. Both of these databases contain a variety of parameters, including information about the experimental conditions. Similarly, SKEMPI (Structural Kinetics and Energetics of Mutant Protein Interactions) is a database of experimentally measured binding free energy changes [18,19]. It includes data for a wide range of protein-protein complexes, and the mutations are annotated with information about their structural and functional effects.

There are numerous computational methods available for predicting the effect of mutations on protein stability and binding [20,21,22,23]. These methods can be broadly divided into two categories: empirical/machine learning (ML) methods and physics-based methods. Empirical methods are based on a statistical analysis of experimental data and use machine learning algorithms to predict the effect of mutations on ΔG_folding_ and ΔG_binding_. Physics-based methods, on the other hand, use principles of thermodynamics and statistical mechanics to predict the effect of mutations on protein stability. However, these methods can account for the complex physical interactions that determine protein stability but require detailed structural information about the protein and are computationally expensive, which makes them non-applicable for genome-scale investigations. In this article, we only deal with fast methods, those using either adjustable parameters or machine learning.

For predicting the effect of mutation on protein stability, several methods have been developed, which can be broadly grouped into structure-based methods and sequence-based methods. The structure-based methods use the protein structure information to derive the features for the wild-type and mutant proteins and then predict the free energy change of the protein due to mutation. The list of the most popular structure-based methods includes FoldX [24], PoPMuSiC [25], mCSM [26], STRUM [27], SDM2 [28], and SAAFEC [29]. The main limitation of these methods is the availability of the 3D structure of the protein of interest. Indeed, only a tiny fraction of known proteins have their 3D structures experimentally determined, which limits the applicability of these methods. This prompted the development of methods that utilize sequence information alone, called sequence-based methods. The most popular include I-Mutant 2.0 [30], Evolutionary, Amino Acid, and Structural Encodings with Multiple Models [31], Impact of Non-synonymous Mutations on Protein Stability [32], BoostDDG [33], and SAAFEC-SEQ [34]. These methods can be applied to genome-scale investigations. Furthermore, it was demonstrated that they outperform some of the structure-based methods despite using only sequence information [34].

The protein–protein binding affinity change of a point mutation has also drawn the attention of the research community. Several computational methods have been reported in the literature for the prediction of binding free energy changes due to point mutations. These methods can be classified into physics-based and knowledge-based methods. Knowledge-based/empirical methods are generally fast and hence better suited for genome-level screening applications like FoldX [24], SAAMBE [35], SAAMBE-3D [36], BindProfX [37], iSEE [38], BeAtMuSiC [39], mCSM-PPI2 [40], and MutaBind2 [41], which require the 3D structure of the complex. In addition, there are a couple of sequence-based methods like SAAMBE-SEQ [42] and ProAffiMuSeq [43], which require sequence only to predict the ΔΔG_binding_ due to the mutation.

Similarly, computational methods for predicting the effect of mutation on protein-nucleic acid ΔG_binding_ have also been developed. The available methods are fewer than the methods for predicting the change in folding or binding free energy of protein–protein interactions, and they all require structural information. The list is quite short and includes FoldX [24], mCSM-NA [44], PremPDI [45], SAMPDI [46], and SAMPDI-3D [47]. It is also to be noted down here that except for SAMPDI-3D, which is a machine learning-based method, all other methods available for prediction are either physics-based or empirical. In addition, only SAMPDI-3D [47] allows the prediction of changes in protein–DNA binding affinity caused by mutations of DNA bases.

The predictions of the effect of mutations on ΔG_folding_ and ΔG_binding_ are essential for protein engineering and understanding the effect of natural variants, i.e., SNVs. We argue that these two tasks may require slightly different approaches and methods. Thus, protein engineering requires methods capable of correctly predicting the effect of any type of mutation on either ΔG_folding_ or ΔG_binding_, with the goal of designing more stable proteins or protein–protein and protein–DNA/RNA complexes with better affinity without any restriction on the type of substitution. In contrast, the methods for predicting ΔG_folding_ and ΔG_binding_ of SNVs focus on mutations seen in nature, i.e., in the human population. The goal of this work is to provide an assessment of leading predictors with respect to predicting the change in ΔG_folding_ and ΔG_binding_ caused by SNV versus non-SNV and how the predictions are affected by the chemical nature of the mutations. It should be mentioned that our investigation sheds light on another aspect of performance assessment, which is different from previous works focusing on the effect of enrichment of destabilizing mutations in the existing experimental databases [48]. Such enrichment was attributed to the less accurate predictions of stabilizing mutations and prompted the creation of balanced datasets [49,50]. Other studies on the performance of the leading algorithms suggested that the problem is overfitting and that the features used in the models are not sufficiently informative for the task [21], as well as the quality of the experimental data [51].

## 2. Results

### 2.1. Assessment of SNV and Non-SNV Energy Change Distribution in Experimental Databases and Types of Amino Acid Changes

Below, we provide the change in the folding and binding free energies distribution in the leading databases for the entire dataset and for SNVs and non-SNVs (Figure 1). It should be mentioned again that the change of the folding free energy vs. the change of the binding free energy is calculated differently (see Equations (1) and (2)). Thus, a negative change in the folding free energy indicates destabilization, while a positive change in the binding free energy points to weaker affinity.

The first observation confirms the previously noticed enrichment of destabilizing mutations, i.e., the vast majority of mutations make the corresponding folding or binding weaker. This is an expected observation since proteins fold toward their lowest folded free energy state. Similarly, the protein-protein and protein–DNA binding free energies are optimized (see [52]). This observation holds for the entire databases and for the subsets, the SNVs and non-SNVs (with a slight exception for protein–DNA databases, the S419 and ProNAB-237 databases, for which the SNV distribution is more symmetrical than that of non-SNVs). Such an imbalance is illustrated in Table 1, where the number of cases with free energy change (positive or negative) larger than 2 kcal/mol and 1 kcal/mol is provided for each of the databases. Indeed, the ratio of destabilizing vs. stabilizing mutations is approximately 8.4 to 16, taking 2 kcal/mol as the cutoff. The ratio is approximately the same (7.7 to 13.7) when the cutoff is decreased to 1 kcal/mol.

In terms of amino acid changes present in the database, the results are provided in the Appendix A. In the S2648 dataset, most of the mutations have been made from Val (319) to other amino acids, followed by Ile (253) and Glu (200) (Appendix A). In terms of mutations, most mutations have been made to Ala, as alanine scanning is one of the most popular methods to study changes in folding free energy as a result of mutation. We see 109 Val-to-Ala mutations and 85 Leu-to-Ala mutations, both of which are SNVs (Appendix A). In case of non-SNVs, we see more cases of Ile to Val mutations (78) and Ala to Gly mutations (75). Considering the properties of amino acids, we see more cases of hydrophobic to hydrophobic mutations, followed by large to small mutations. SNV cases are dominated by hydrophobic to hydrophobic and small to small mutations, while non-SNV cases are dominated by large to small and polar to hydrophobic mutations (Appendix A).

The trend of amino acid change is slightly different in the SKEMPI-SEQ-2388 (Appendix A) and SKEMPI-3D-3775 datasets (Appendix A). While we see more cases where Arg (238) is mutated to other amino acids, followed by Glu (235) and Lys (217), in the SKEMPI-SEQ-2388 dataset, the SKEMPI-3D-3775 dataset has more cases of Lys (354) and Arg (352) mutations to other amino acids, followed by glutamic acid (320) (Appendix A). Again, most of the amino acids have been mutated to Ala, with Glu-to-Ala mutations (cases of SNV) being the most prominent ones in both datasets. With reference to the properties of amino acids, most of the mutations have been made from polar to hydrophobic and large to small in both datasets. Both datasets are dominated by non-SNV mutations from large to small amino acids (Appendix A).

Similar to the S2648 and two SKEMPI datasets, the datasets used for studying the effect of mutation on protein-DNA binding free energy are also dominated by mutations to alanine (Appendix A). In both the datasets (S419 and ProNAB-237), most of the mutations have been made from Arg to other amino acids (S419: 76 and ProNAB-237: 57), with Arg to Ala being the prominent ones (Appendix A). We see more cases of large to small and polar to hydrophobic mutations in the S419 and ProNAB-237 datasets (Appendix A).

### 2.2. Assessment of Leading Algorithms Sensitivity with Regard to SNV and Non-SNV Mutations

In this section, we benchmark the leading free energy change predictors against the datasets listed above. It must be clarified that the benchmarking is carried out with the sole purpose of revealing the difference in performance between SNVs and non-SNV cases. It is understood that these algorithms were trained on the above-listed databases, and thus their absolute performance should not be evaluated on the same datasets.

Table 2 shows the Person correlation coefficient (PCC) and mean squared error (MSE) (defined in Equation (3) and Equation (4), respectively) of predictions of folding free energy changes (the linear fit graphs of predicted vs. experimentally measured folding free energy changes are shown in Appendix A). Two observations can be made: almost all algorithms perform better on non-SNV cases compared with the whole dataset, and almost all algorithms perform much better on non-SNVs. The largest difference between SNVs and non-SNVs was found for MAESTRO, mCSM, PoPMuSiC, and DUET. In contrast, the MSE is worse for non-SNVs compared with SNV cases. The best slope of the fitting lines was obtained in the case of SAAFEC-SEQ, and there is a practically very minor difference in the performance of SAAFEC-SEQ on the whole dataset, SNV and non-SNV. The largest difference in the slope of the fitting lines was seen in the cases of MAESTRO, DUET, mCSM, and PoPMuSiC when comparing SNV and non-SNV cases. The average slope obtained in the case of the whole dataset was 0.48 for SNVs and 0.4 for non-SNVs, it was 0.54, again demonstrating that the method works better on non-SNVs. It is interesting to mention that the slope of the fitting line for all algorithms is about 0.5, indicating that they underpredict the change of the folding free energy by a factor of 2.

Table 3 summarizes the results for protein-protein binding free energy change predictors (the linear fit graphs of predicted vs. experimentally measured binding free energy changes are shown in Appendix A. Similarly, as above, two observations can be made: all algorithms (except for MutaBind2) perform slightly better on non-SNV cases compared with the whole dataset, and all algorithms perform better on non-SNVs compared with SNVs (except MutaBind2). However, the differences in PCC are not as large as they were in the case of folding free energy change predictors. The slope of the fitting lines is nearly identical for SNV and non-SNV in the cases of SAAMBE-3D and MutaBind2 (Table 3); however, we see a better slope for non-SNV in the case of mCSM-PPI2 (SNV = 0.60 and non-SNV = 0.68). The slope of BeAtMuSiC is very small, reaching 0.13 for SNV cases, indicating that the predicted binding free energy changes are grossly underestimated. All other methods also underestimate the change in binding free energy by about a factor of 1.5.

Lastly, Table 4 provides PCC and MSE for protein-DNA binding free energy change predictors (the linear fit graphs of predicted vs. experimentally measured binding free energy changes are shown in Appendix A. The mCSM-NA results indicate that it performs better on SNV cases, while both SAMPDI-3D and PremPDI have almost identical performance on SNV and non-SNV cases on the S419 dataset. The corresponding MSEs for SNVs and non-SNVs are almost identical for SAMPDI-3D and opposite in tendency for the other two algorithms. It should be noted that the slopes of fitting lines in the case of mCSM-NA are very low for both SNVs and non-SNVs, underestimating the binding free energy change by a factor of 4. Similarly, the PremPDI predictions on the SNVs case are underestimated by a factor of 3. The performance is the opposite in the case of the ProNAB-237 dataset. All three methods perform better on non-SNVs as compared to SNVs. The slopes of the fitting lines are not impressive, resulting in an underestimation of a factor of 3 to 4.

### 2.3. Assessment of Leading Algorithms Sensitivity Based on the Chemical Nature of Amino Acid Mutations

It is to be noted that all the methods for predicting changes in folding free energy or change in binding free energy (protein–protein or protein–DNA) were developed and optimized against a broad range of mutations. Therefore, in this section, we wanted to understand how these methods perform when different types of amino acid mutations/are considered. The following kinds of mutations were considered based on the properties of amino acids: hydrophobic–hydrophobic, hydrophobic–polar, polar–polar, polar–hydrophobic, small–small, small–large, large–large, large–small, aliphatic–aliphatic, aliphatic–aromatic, aromatic–aromatic, aromatic–aliphatic, positive–positive, positive–negative, negative–negative, and negative–positive. In the case of the S2648, SKEMPI-SEQ-2388, and SKEMPI-3D-3775 datasets, only those categories were considered for analysis where the number of cases was more than 10% of the total dataset (for example, the S2648 dataset has 2648 data points, and 10% of them is 265). Thus, if a given subset had fewer than 265 data points, it was not considered. A different threshold was applied to protein–DNA databases since, in the case of S419 and ProNAB-237, the total number of cases is relatively small compared to other datasets, forcing us to apply a 50% cut-off of the total dataset (Thus, for S419, the 50% threshold results in 210 datapoints, and if any of the subclasses had less than 210 entries, it was not considered in the analysis). The results for the performance of change in folding free energy predictors, change in protein-protein binding free energy predictors, and change in protein-DNA binding free energy predictors are shown in Table 5, Table 6 and Table 7.

In the case of changes in folding free energy predictors, SAAFEC-SEQ consistently results in PCC ranging from 0.88 to 0.92 (Table 5) and is comparable to its performance on the total dataset. Except for polar-polar, polar-hydrophobic, and large-large, the slope of the fitting line is also nearly the same for all classes. It is interesting to note that nearly all methods perform relatively better on aliphatic-aliphatic substitutions and worse on polar-hydrophobic substitutions compared to the total dataset. There are cases where the method performs better on some classes compared to the total dataset, and vice versa. For instance, the performance of I-mutant 2.0 (sequence-based) performs better on aliphatic-aliphatic (PCC: 0.62) compared to the total dataset (PCC: 0.55). The performance of SDM is worse in the case of large–large substitutions (PCC: 0.27) compared to its performance on the total dataset (0.46). Overall, we do not observe bias in the performance of SAAFEQ-SEQ on a particular subset; however, other methods show bias in their performance on different datasets, making their performance go from better to worse.

Table 6 shows the performance of various predictors of protein-protein binding free energy change for the various subsets mentioned above. Several observations can be made: BeAtMuSiC outperforms by a factor of 1.4 in the cases of hydrophobic–hydrophobic and large–small mutations compared to its performance on the total dataset and other classes. However, its performance is worse in the case of small–large (PCC: 0.13) and large–large (PCC: 0.19) mutations. SAAMBE-SEQ performs worse in cases of hydrophobic–hydrophobic mutations (PCC: 0.80) and SAAMBE-3D in cases of small–small mutations (PCC: 0.84). Similarly, the cases with MutaBind2 and mCSM-PPI2.

Table 7 shows the results for the predictors of protein-DNA change in binding free energy for the subsets. For the S419 dataset, the performance of all the methods was worse for the subsets compared to the total dataset. For ProNAB-237, the performance of SAAMPDI-3D was slightly better for the subsets, and mCSM-NA performed better in cases of large–small mutations.

## 3. Discussion

This article aimed to reveal the differences between SNVs and non-SNVs in terms of their distributions in the corresponding databases and the performance of leading algorithms for free energy change predictions. Three types of databases were considered: the folding free energy changes, the protein–protein binding free energy changes, and the protein–DNA binding free energy changes. It should be mentioned that the first two are much larger than the third one, and therefore the observations made are statistically more meaningful for the first two. The common observation is that SNVs and non-SNVs are almost equally represented in the databases; roughly speaking, 50% are SNVs and 50% are non-SNVs. The corresponding free energy changes, ΔG_folding_ and ΔG_binding_, are similar as well, except for protein–DNA databases, where ΔG_binding_ of SNVs does not have as many destabilizing cases as non-SNVs do. The main difference between SNVs and non-SNVs in the corresponding databases is the type of mutation. For instance, SNV cases in S2648 are dominated by hydrophobic to hydrophobic and small to small mutations, while non-SNV cases are dominated by large to small and polar to hydrophobic mutations, whereas SNV cases in both SKEMPI-SEQ-2388 and SKEMPI-3D-3775 are dominated by small–small and polar to hydrophobic mutations, while non-SNV mutations are from large to small amino acids. We see more cases of polar to hydrophobic and small to small mutations in the case of SNV in the S419 and ProNAB-237 datasets, and non-SNV by large to small and polar to hydrophobic mutations. These differences between SNVs and non-SNVs should be taken into consideration when selecting features for machine learning algorithms for predicting the effects of SNVs. Alternatively, if the goal is to develop a method that predicts the effects of SNVs only, only SNV cases should be used for the training set.

In terms of the performance of the leading predictors of free energy change, ΔG_folding,_ and ΔG_binding_, we would like to reiterate again that our goal is not to compare their absolute performance but rather to see the difference in the performance of SNVs vs. non-SNVs cases. A comparison of their performance has been carried out in numerous papers of the developers [26,28,30,32,34,36,39,40,41,42,44,45,47,53,54,55,56], as well as in third-party manuscripts [57]. The common observation is that almost all algorithms tested on the corresponding datasets perform worse on SNVs as compared with non-SNVs. In some cases, the PCC for SNVs is two times lower than the PCC for non-SNVs. This observation should be considered when asserting the effect of SNVs, both benign and pathogenic, on protein stability and macromolecular interactions. Especially since there is a strong linkage between thermodynamics and pathogenicity [58].

The second main goal of this investigation was to probe the sensitivity of the leading predictors with respect to the chemical nature of mutations. This was motivated by the fact that some mutations involve a drastic change in the chemical nature of the mutation site (from hydrophobic to charged amino acid), while others may preserve the overall chemical nature of the site (from hydrophobic to hydrophobic). An algorithm that properly uses such information in the corresponding features is expected to perform equally on such chemically distinctive cases, while if such properties were not used in the algorithm, one should observe different performance (in terms of PCC and MSE). Indeed, as shown above (Table 5, Table 6 and Table 7), some algorithms are almost insensitive to the types of mutations and can be used in future investigations without concern about the chemical nature of the mutation. However, other algorithms demonstrated significant sensitivity to the chemical nature of the mutation, and they should be used with caution.

## 4. Materials and Methods

### 4.1. Databases of Experimentally Measured Changes of ΔG_folding_ and ΔG_binding_

#### 4.1.1. Databases of Folding Free Energy Changes (Main Source ProTherm Database)

ProTherm is a database of experimentally measured thermodynamic quantities for wild-type and mutant proteins [14,15]. The database includes information about protein stability in the form of unfolding free energy and thus provides important clues about stability and its association with structure as a result of mutation. This database has been widely utilized as a training dataset for developing in silico methods for the prediction of changes in folding free energy as an effect of the mutation. It is to be noted down here that the majority of the methods are trained on the subset of the ProTherm database named the S2648 dataset.

***S2648:*** The S2648 dataset, originally curated by Dehouck et al. [55], comprises 2648 single-point mutations from 131 proteins. The criteria used for defining the dataset are described in the article by Dehouck et al. [55]. For the current work, we used S2648 for benchmarking purposes. One of the structures (PDB Id: 2A01) is now marked as obsolete, which corresponds to one entry in the dataset that was removed. The final S2648 dataset used in the current work consists of 2647 single-point mutations from 130 proteins.

#### 4.1.2. Databases of Binding Free Energy Changes of Protein-Protein Interactions (Main Source SKEMPI Database)

Using the Structural Kinetic and Energetic database of Mutant Protein Interactions (SKEMPI) [18] as the main source of information, two datasets were created to benchmark the methods for computing the change in ΔG_binding_ as a result of mutation from the second version of the SKEMPI database called SKEMPI-2.0 [19]. First, to benchmark the sequence-based methods, and second, to benchmark the structure-based methods. We named these datasets SKEMPI-SEQ-2388 (sequence information only) and SKEMPI-3D-3775 (SKEMPI entries with the corresponding 3D structures). Both of these datasets were built by considering single-point mutations only. The criteria used for curating the two datasets are described below:

***SKEMPI-SEQ-2388*:** The SKEMPI-SEQ-2388 dataset is a curated subset of the SKEMPI-2.0 database, which has 7085 mutations from 348 protein–protein complexes. The curation starts with purging entries that have a missing temperature or an exact *K*_d_ value for wild-type or mutants, followed by considering only single point mutations, resulting in 4827 mutations over 316 protein complexes. Subsequently, 3000 mutant cases from 210 protein-protein complexes involving only dimers were considered, and multimers were filtered out. Then we calculated the average and standard deviation of binding free energy changes for all the mutations with multiple experimental values in the dataset. We removed all duplicated mutations if the corresponding ΔΔG_binding_ had a standard deviation greater than 1 kcal/mol and collapsed the rest into a single entry if the standard deviation was less than 1 kcal/mol, resulting in 2450 mutations and over 210 complexes. Finally, we filtered out cases where any of the involved protein chains have less than 20 amino acids, resulting in a dataset of 2388 mutations from 200 protein-protein complexes.

***SKEMPI-3D-3775*:** SKEMPI-3D-3775 is also a curated subset of the SKEMPI-2.0 database. The first curation step is identical to that for the SKEMPI-SEQ-2388 dataset, and it resulted in 4827 mutations in 316 proteins. In mutation entries where multiple experimental values are listed, the mean and standard deviation of the ΔΔG_binding_ for the mutation are computed, and if the standard deviation of ΔΔG_binding_ is greater than 1.0 kcal/mol, the mutant data point is eliminated; otherwise, we consider the average ΔΔG_binding_ for the mutation, which will be included in the cleaned dataset only once. After this cleaning step, we obtained 4067 mutations from 316 protein–protein complexes. Afterward, cases with missing residues in the −5 to +5 position of the mutation site were also discarded, as this information is required by some of the methods [36]. Some of the structures have non-standard amino acid analogs; such amino acids were reverted to their respective parent amino acids. Structures containing non-standard residues with RCSB PDB chemical IDs CGU or LLP were purged from the dataset. Finally, our curated dataset consists of 3775 mutations from 300 protein–protein complexes.

#### 4.1.3. Databases of Binding Free Energy Changes of Protein–DNA Interactions

ProNIT [16] and ProNAB [17] are the two databases that contain information about the experimental binding data for protein-nucleic acid complexes. ProNIT release 2.0 [59] contains 4900 from 158 proteins. The database contains information about several thermodynamic quantities like Gibbs free energy change (ΔG), enthalpy change (ΔH), dissociation constant (Kd), association constant (Ka), heat capacity change (ΔCp), structural information of the protein–nucleic acid complexes, and other information like experimental conditions, literature information, etc. The database is currently not active. ProNAB (Harini et al., 2022 [17]) is a database of experimentally measured protein-nucleic acid binding affinities of wild-type and mutant proteins. The database is cross-linked with multiple databases like UniProt, GenBank, PDB, PROSITE, ProThermDB, DisProt, and PubMed. The current version of ProNAB consists of 20,219 entries from 1041 proteins, which include 14,732 cases of Protein–DNA, 5326 cases of Protein–RNA, and 161 cases with Protein–DNA/RNA hybrid binding affinity. We used two datasets for benchmarking purposes, which are described below.

***S419:*** This dataset has been curated by Gen Li et al. [47] and has been used as the training set for SAMPDI-3D [47]. The S419 dataset comprises 419 single-point mutations in 96 proteins. The dataset was created by merging two datasets: (a) the S219 dataset, collected from ProNIT and dbAMEPNI, which was used as the training set for the development of PremPDI [45], and (b) the S200 dataset, collected from the recently published literature [47].

***ProNAB-237:*** ProNAB-237 is a subset of the ProNAB database [17] used in the current study. ProNAB includes cases of both single-point mutations and multiple mutations. We only considered single-point mutations for the current work. We collected 4806 cases of nucleic acid–protein-free binding energies with a single amino acid substitution. The dataset was filtered to exclude cases where the nucleic acid type was either RNA or another. This step resulted in 860 data points. The dataset was further filtered to remove cases where PDB structure was not reported for the protein-DNA complex, resulting in 631 cases from 137 proteins. For cases where multiple measurements were reported for the same mutation, the standard deviation was calculated for the changes in the binding affinity, and only those cases were considered where the standard deviation was less than 1.0 kcal·mol^−1^ for a given mutation. In these cases, the average value of the change in binding free energy was taken into account and used for benchmarking. The dataset was further pruned to remove mutations for which atomic coordinates were absent in the PDB. In some PDBs, the mutation residue listed in the database does not map to the residue in it; such cases were also removed, and finally, we have a dataset of 237 mutations. We named this dataset ProNAB-237.

### 4.2. Computational Methods for Predicting ΔG_folding_ or ΔG_binding_

#### 4.2.1. Methods for Predicting Folding Free Energy Change Caused by Mutation

Altogether, eight different methods were used for the benchmarking, three of which are sequence-based, while others require 3D structure as the input for making predictions. Below, we briefly outline these methods. While there are plenty of methods available for the prediction, the following were chosen because they are popular, easy to use, and easy to install:SAAFEC-SEQ [34]: SAAFEC-SEQ is a gradient-boosting decision tree machine learning method that uses physicochemical properties, sequence features, and evolutionary information features to predict changes in folding free energy caused by amino acid mutation. The method utilizes amino acid sequences as input for making predictions;INPS-MD [32,53]: INPS-MD has been implemented as both a sequence (INPS) and a structure-based method (INSPS-3D). Both are machine learning methods based on support vector regression (SVR);I-mutant 2.0 [30]: I-mutant 2.0 is a support vector machine (SVM)-based method for prediction of folding free energy as an effect of mutation. The method is implemented as both sequence- and structure-based;mCSM [26]: mCSM is a web-based predictor that uses a graph-based approach to predict the impact of missense mutations on protein stability. The predictive models in mCSM are trained with the atomic distance patterns of different amino acid residues;MAESTRO [54]: MAESTRO is a structure-based method that utilizes a multi-agent machine learning system for predicting the impact of mutation on folding free energy;PoPMuSic [60]: PoPMuSiC is a web server that predicts the thermodynamic stability changes caused by single-site mutations in proteins, using a linear combination of statistical potentials whose coefficients depend on the solvent accessibility of the mutated residue;SDM [61]: Site-Directed Mutator (SDM) uses the statistical potential energy function to calculate the stability score, which uses amino-acid substitution frequencies within homologous protein families. The metric is analogous to the free energy difference between wild-type and mutant proteins. The method is 3D structure based and is available as a webserver;DUET [56]: DUET is a 3D structure-based method that uses mCSM and SDM for consensus prediction. The results from these methods are combined and optimized using Support Vector Machines (SVM) to make the final prediction. The method is available as a web server.

#### 4.2.2. Methods for Predicting Binding Free Energy Changes of Protein–Protein Interactions Caused by Mutation

Overall, the following listed computational methods were considered in this work. We could not include methods that are designed for predicting the effects of single amino acid mutations on binding free energy change for dimeric complexes or whose online servers were slow or busy and stand-alone versions were not available or not trivial to install and configure locally.

SAAMBE-SEQ [42]: It is a sequence-based machine-learning technique that can predict how a single mutation will affect the binding energy of protein–protein complexes. In contrast to other methods already in use, SAAMBE-SEQ does not require a 3D protein–protein complex structure as input. Note that it uses features that require the length of interacting partners to be longer than 20 amino acids, and thus it is not expected to perform well on protein–peptide binding cases;SAAMBE-3D [36]: SAAMBE-3D is a machine learning-based method that takes a PDB file as its input and can estimate the effect of a single amino acid modification on protein-protein binding. This tool enables the investigation of two types of inquiries: (1) forecasting alterations in binding free energy resulting from a mutation; and (2) predicting whether a mutation causes a disturbance in protein–protein interactions;mCSM-PPI2 [40]: mCSM-PPI2 is a computational technique that uses machine learning to forecast the impact of missense mutations on protein-protein binding affinity. It employs an enhanced graph-based signature strategy to model changes in the network of non-covalent interactions between residues using graph kernels, complex network metrics, evolutionary data, and energetic terms. This approach is available for free at https://biosig.lab.uq.edu.au/mcsm_ppi2/ (accessed on 1 May 2023);MutaBind2 [41]: MutaBind2 is a tool that assesses the influence of individual-site and multi-site mutations on protein-protein binding affinities in soluble complexes. This method utilizes statistical potentials, molecular mechanics, force fields, and the structure of the protein-protein complex;BeAtMuSiC [39]: BeAtMuSiC is a method based on a set of statistical potentials derived from known protein structures. In addition, it accounts for the effect of the mutation on the strength of the interactions at the interface as well as the overall stability of the complex. This method is available as an online web server free of charge at http://babylone.3bio.ulb.ac.be/beatmusic/index.php (accessed on 1 May 2023).

#### 4.2.3. Methods for Predicting Binding Free Energy Changes of Protein–DNA Interactions Caused by Mutation

Three methods were utilized for benchmarking the binding free energy changes of protein–DNA interactions in the presence of a chemical modification. Both methods rely on a three-dimensional structure for the prediction of the binding free energy of protein–DNA interactions.

SAMPDI-3D [47]: SAMPDI-3D uses a gradient-boosting decision tree machine learning method to predict the change in the protein–DNA binding free energy brought on by mutations in the binding protein or the bases of the corresponding DNA. It takes the structure of the complex, i.e., a PDB file, as an input;mCSM-NA [44]: The mCSM-NA method is based on graph-based structural signatures to predict the DDG caused by mutations in proteins bound to DNA/RNA;PREMPDI [45]: PremPDI is a physics-based method that relies on the 3D structure of the protein–nucleic acid complex for making predictions. The method is based on molecular mechanic force fields and fast side-chain optimization algorithms.

### 4.3. SNV vs. Non-SNV Cases

The SNV cases were extracted from the corresponding experimental databases using the lookup table provided in the Supplementary Material (Appendix A). The rest of the cases were considered non-SNV. In the S2648 dataset, there are 1493 cases of SNVs and 1154 cases of non-SNVs (Appendix A). In the case of SKEMPI-SEQ-2388 and SKEMPI-3D-3775, there are 1081 and 1692 SNVs and 1307 and 2083 non-SNVs, respectively (Appendix A). Similarly, for S419 and ProNAB-237, there are 164 and 79 cases of SNVs and 255 and 158 cases of non-SNVs (Appendix A). It is to be noted that the cases of SNVs and non-SNVs are not equal in all the datasets. Except for S2648, where cases of SNVs are higher compared to non-SNVs, all other datasets have more cases of non-SNVs.

### 4.4. Free Energy Changes

Following previous papers, the change of the folding free energy is calculated as the change of the folding or binding free energy produced by a single amino acid substitution as the difference of the folding and binding free energies of the wild type and the mutant [42,62]. However, the corresponding free energy changes are calculated differently for folding free energy changes versus binding free energy changes. Thus, the change in folding free energy caused by a mutation is calculated with Equation (1), and a positive ΔΔG indicates a mutation that makes the protein stable, while a negative value is representative of destabilization.
ΔΔG_folding_ = ΔG_wt_ − ΔG_mutant_
(1)

In contrast, the change in binding free energy for both protein–protein and protein–DNA complexes was calculated as
ΔΔG_binding_ = ΔG_mut_ − ΔG_wt_(2)
and thus, a positive number indicates that the mutation destabilizes binding, while a negative number indicates that the mutation makes the affinity stronger.

### 4.5. Sampling and Assessment of Predictions

To avoid a plausible bias toward the selection of data points, we randomly pick up 100 times “N” number of datapoints, where “N” is defined as the minimum of 50% of the SNV or non-SNV cases. For example, in the S2648 dataset, there are 1493 cases of SNVs and 1154 cases of non-SNVs. We pick up the smaller number, 1154 non-SNV cases, and 50% is 572 cases. Thus, 572 cases from the whole S2648 dataset and from subsets of SNVs and non-SNVs are randomly selected 100 times. Then we report the averaged PCC, MSE, and standard deviation. For the analysis of chemically different types of mutations, i.e., hydrophobic to hydrophobic, etc., we first check if the total number of datapoints is greater than 10% of all available data points. If not, the class is not considered. For example, 10% of S2648 dataset is 265 data points (Exceptions to this rule are the S419 and ProNAB-237 databases, because the total number of cases is relatively small compared to other datasets, which made us consider only categories where the total number of cases is at least equal to 50% of the total dataset). Then we randomly pick up 100 times “K” number of datapoints from the corresponding dataset, where “K” is defined as 50% of the data points (for example, the number of cases in the hydrophobic-to-hydrophobic dataset, etc.). Then we report the averaged PCC, MSE, and standard deviation.

The accuracy of the predictions was assessed using two measures, namely the Pearson correlation coefficient (PCC) and the mean square error (MSE), which are defined below.
(3)$$PCC=\frac{\sum_{i=1}^{n}\left(x_{i}-\bar{x})(y_{i}-\bar{y}\right)}{\sqrt{\sum_{i=1}^{n}\;{\left(x_{i}-\bar{x}\right)}^{2}{\left(y_{i}-\bar{y}\right)}^{2}}}\;$$
where $x_{i}$ and $y_{i}$ refer to the true and predicted values of the *i*th sample.
(4)$$MSE\left(y,\hat{y}\right)=\frac{1}{n_{samples}}\sum_{i=0}^{n_{samples}-1}{\left(y-\hat{y}\right)}^{2}$$
where $y_{i}$ is the true value and ${\hat{y}}_{i}$ is the predicted value of the *i*th sample.

## 5. Conclusions

The paper showed that the distribution of SNV vs. non-SNV types of mutations is different in the corresponding databases of experimentally measured quantities. This should be taken into consideration when one applies methods of predicting the effect of missense mutations seen in the human population. Furthermore, the work indicated that the leading algorithms for predicting folding and binding free energy changes caused by mutations perform differently in cases of SNVs and non-SNVs. This is an important observation since there is a strong linkage between the change in folding and binding free energies and the probability of mutation being pathogenic [58]. Overall, PCC is better for non-SNVs, which points out that the methods may not be accurate in ranking SNV cases. In contrast, in terms of MSE, most methods have a larger MSE for non-SNV cases, which may indicate that they are more accurate in predicting individual energy changes for non-SNVs. However, this comes with an overall worsening slope of the fitting line, which indicates an underestimation of the energy change. Furthermore, it was shown that some of the leading algorithms are very sensitive with respect to the chemical nature of the mutation, which should also be taken into consideration when they are applied to future investigations, especially cases involving mutations in Ala.

## Figures and Tables

**Figure 1 ijms-24-12073-f001:**
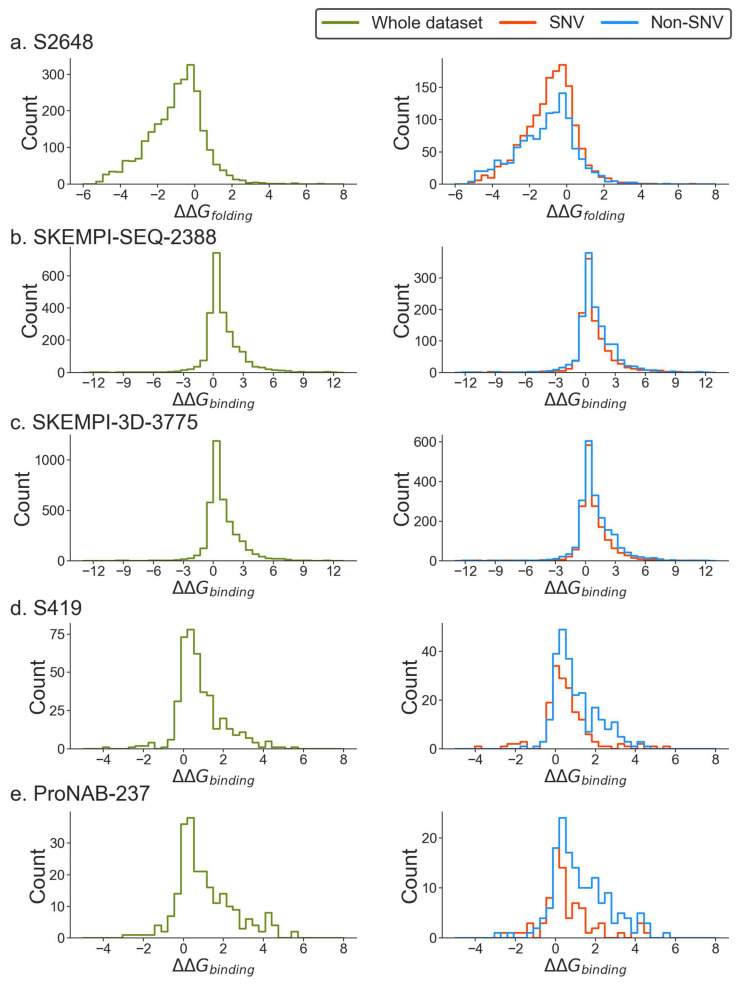
Distribution of change in folding and binding free energy for different datasets.

**Table 1 ijms-24-12073-t001:** Total number of stabilizing and destabilizing mutations in the datasets.

Datasets	Cut-Off			
	>2 kcal/mol		>1 kcal/mol	
	No. of Stabilizing Mutations	No. of Destabilizing Mutations	No. of Stabilizing Mutations	No. of Destabilizing Mutations
S2648	42	621	152	1192
SKEMPI-SEQ-2388	56	470	117	903
SKEMPI-3D-3775	67	742	159	1398
S419	4	64	10	137
ProNAB-237	3	53	9	100

**Table 2 ijms-24-12073-t002:** Sensitivity comparison of methods on the S2648 dataset with regard to SNV and non-SNV mutations. The table provides the PCC, MSE, and slope of the fitting line, along with the standard deviation of each of them taken over 100 random selections of data points (see Section 4 for details).

Methods	S2648								
	Whole Dataset			SNV			Non-SNV		
	PCC	MSE	Slope	PCC	MSE	Slope	PCC	MSE	Slope
SAAFEC-SEQ ^a^	0.91 ± 0.01	0.46 ± 0.03	0.67 ± 0.01	0.9 ± 0.01	0.45 ± 0.04	0.64 ± 0.02	0.92 ± 0.01	0.47 ± 0.03	0.69 ± 0.01
I-mutant 2.0 ^a^	0.55 ± 0.02	1.71 ± 0.07	0.46 ± 0.02	0.52 ± 0.03	1.49 ± 0.09	0.38 ± 0.03	0.57 ± 0.02	1.94 ± 0.1	0.52 ± 0.02
I-mutant 2.0	0.61 ± 0.02	1.53 ± 0.08	0.52 ± 0.02	0.57 ± 0.03	1.39 ± 0.11	0.44 ± 0.04	0.63 ± 0.02	1.67 ± 0.09	0.57 ± 0.02
INPS ^a^	0.58 ± 0.02	1.58 ± 0.06	0.4 ± 0.01	0.53 ± 0.02	1.5 ± 0.1	0.34 ± 0.02	0.6 ± 0.02	1.66 ± 0.08	0.43 ± 0.02
INPS-3D	0.65 ± 0.01	1.31 ± 0.06	0.42 ± 0.01	0.59 ± 0.02	1.28 ± 0.1	0.35 ± 0.02	0.68 ± 0.02	1.34 ± 0.07	0.46 ± 0.02
mCSM	0.69 ± 0.01	1.16 ± 0.05	0.43 ± 0.01	0.62 ± 0.02	1.19 ± 0.08	0.34 ± 0.02	0.74 ± 0.02	1.12 ± 0.06	0.5 ± 0.01
MAESTRO	0.66 ± 0.01	1.29 ± 0.05	0.54 ± 0.02	0.59 ± 0.02	1.31 ± 0.08	0.43 ± 0.02	0.72 ± 0.02	1.26 ± 0.07	0.61 ± 0.02
PoPMuSiC	0.62 ± 0.02	1.36 ± 0.06	0.41 ± 0.01	0.55 ± 0.03	1.33 ± 0.1	0.33 ± 0.02	0.67 ± 0.02	1.38 ± 0.08	0.48 ± 0.02
SDM	0.46 ± 0.02	2.38 ± 0.08	0.44 ± 0.02	0.4 ± 0.02	2.16 ± 0.12	0.36 ± 0.03	0.5 ± 0.02	2.61 ± 0.11	0.5 ± 0.03
DUET	0.69 ± 0.01	1.17 ± 0.05	0.51 ± 0.01	0.62 ± 0.02	1.19 ± 0.08	0.41 ± 0.02	0.73 ± 0.02	1.16 ± 0.06	0.59 ± 0.02

^a^ Sequence-based method.

**Table 3 ijms-24-12073-t003:** Sensitivity comparison of methods on the SKEMPI-SEQ-2388 and SKEMPI-3D-3775 datasets with regard to SNV and non-SNV mutations. The table provides the PCC, MSE, and slope of the fitting line, along with the standard deviation of each of them taken over 100 random selections of data points (see Section 4 for details).

Methods	SKEMPI-SEQ-2388								
	Whole Dataset			SNV			Non-SNV		
	PCC	MSE	Slope	PCC	MSE	Slope	PCC	MSE	Slope
SAAMBE-SEQ	0.88 ± 0.01	0.83 ± 0.08	0.71 ± 0.02	0.87 ± 0.02	0.87 ± 0.11	0.69 ± 0.03	0.89 ± 0.01	0.79 ± 0.11	0.73 ± 0.03
	**SKEMPI-3D-3775**								
	**Whole Dataset**			**SNV**			**Non-SNV**		
	**PCC**	**MSE**	**Slope**	**PCC**	**MSE**	**Slope**	**PCC**	**MSE**	**Slope**
SAAMBE 3D	0.9 ± 0.01	0.67 ± 0.04	0.64 ± 0.01	0.89 ± 0.01	0.65 ± 0.06	0.63 ± 0.02	0.91 ± 0.01	0.68 ± 0.06	0.65 ± 0.01
MutaBind2	0.9 ± 0.01	0.61 ± 0.03	0.7 ± 0.01	0.9 ± 0.01	0.58 ± 0.05	0.69 ± 0.02	0.9 ± 0.01	0.65 ± 0.04	0.7 ± 0.01
mCSM-PPI2	0.91 ± 0.01	0.66 ± 0.05	0.64 ± 0.01	0.88 ± 0.02	0.75 ± 0.08	0.6 ± 0.02	0.93 ± 0.01	0.57 ± 0.06	0.68 ± 0.01
BeAtMuSiC	0.34 ± 0.03	2.74 ± 0.19	0.18 ± 0.02	0.31 ± 0.04	2.54 ± 0.25	0.13 ± 0.02	0.37 ± 0.04	2.95 ± 0.29	0.21 ± 0.03

**Table 4 ijms-24-12073-t004:** Sensitivity comparison of methods on S419 and ProNAB-237 datasets with regard to SNV and non-SNV mutations. The table provides the PCC, MSE, and slope of the fitting line, along with the standard deviation of each of them taken over 100 random selections of data points (see Section 4 for details).

Methods	S419								
	Whole Dataset			SNV			Non-SNV		
	PCC	MSE	Slope	PCC	MSE	Slope	PCC	MSE	Slope
SAMPDI-3D	0.84 ± 0.02	0.47 ± 0.05	0.53 ± 0.03	0.84 ± 0.03	0.49 ± 0.08	0.52 ± 0.04	0.8 ± 0.04	0.45 ± 0.07	0.52 ± 0.04
mCSM-NA	0.38 ± 0.07	1.56 ± 0.17	0.33 ± 0.06	0.41 ± 0.1	1.48 ± 0.24	0.35 ± 0.1	0.27 ± 0.1	1.64 ± 0.24	0.24 ± 0.08
PremPDI	0.45 ± 0.06	1.49 ± 0.24	0.43 ± 0.06	0.4 ± 0.08	1.34 ± 0.23	0.3 ± 0.06	0.43 ± 0.07	1.64 ± 0.42	0.48 ± 0.11
	**ProNAB-237**								
	**Whole Dataset**			**SNV**			**Non-SNV**		
	**PCC**	**MSE**	**Slope**	**PCC**	**MSE**	**Slope**	**PCC**	**MSE**	**Slope**
SAMPDI-3D	0.57 ± 0.07	1.43 ± 0.21	0.29 ± 0.04	0.46 ± 0.14	1.58 ± 0.29	0.2 ± 0.06	0.59 ± 0.11	1.29 ± 0.32	0.31 ± 0.07
mCSM-NA	0.4 ± 0.1	2.33 ± 0.42	0.34 ± 0.08	0.28 ± 0.13	3.23 ± 0.79	0.28 ± 0.13	0.55 ± 0.12	1.43 ± 0.33	0.4 ± 0.09
PremPDI	0.53 ± 0.06	1.78 ± 0.27	0.45 ± 0.08	0.45 ± 0.08	2.11 ± 0.45	0.39 ± 0.09	0.54 ± 0.11	1.46 ± 0.33	0.39 ± 0.09

**Table 5 ijms-24-12073-t005:** Sensitivity comparison of methods on the S2648 dataset according to chemically different types of amino acid mutation. The table provides PCC, MSE, and slope of the fitting line, along with the standard deviation of each of them taken over 100 random selections of data points (see Section 4 for details).

Methods	Types of Amino Acid Mutation	S2648		
		PCC	MSE	Slope
SAAFEC-SEQ	Hydrophobic–Hydrophobic	0.91 ± 0.02	0.44 ± 0.03	0.67 ± 0.01
	Hydrophobic–Polar	0.91 ± 0.02	0.44 ± 0.08	0.67 ± 0.03
	Polar–Polar	0.90 ± 0.01	0.46 ± 0.06	0.61 ± 0.02
	Polar–Hydrophobic	0.89 ± 0.01	0.47 ± 0.04	0.60 ± 0.02
	Small–Small	0.91 ± 0.01	0.46 ± 0.04	0.65 ± 0.02
	Small–Large	0.90 ± 0.02	0.38 ± 0.05	0.65 ± 0.02
	Large–Large	0.88 ± 0.02	0.50 ± 0.06	0.62 ± 0.02
	Large–Small	0.91 ± 0.01	0.44 ± 0.03	0.68 ± 0.01
	Aliphatic–Aliphatic	0.92 ± 0.01	0.35 ± 0.03	0.71 ± 0.01
I-mutant 2.0 ^a^	Hydrophobic–Hydrophobic	0.53 ± 0.02	1.74 ± 0.08	0.44 ± 0.02
	Hydrophobic–Polar	0.45 ± 0.06	1.97 ± 0.25	0.34 ± 0.05
	Polar–Polar	0.60 ± 0.06	1.24 ± 0.11	0.41 ± 0.05
	Polar–Hydrophobic	0.42 ± 0.03	1.74 ± 0.11	0.33 ± 0.03
	Small–Small	0.45 ± 0.03	1.83 ± 0.11	0.32 ± 0.03
	Small–Large	0.55 ± 0.04	1.32 ± 0.13	0.43 ± 0.05
	Large–Large	0.47 ± 0.05	1.70 ± 0.12	0.36 ± 0.04
	Large–Small	0.58 ± 0.03	1.68 ± 0.11	0.53 ± 0.03
	Aliphatic–Aliphatic	0.62 ± 0.03	1.37 ± 0.09	0.53 ± 0.03
I-mutant 2.0	Hydrophobic–Hydrophobic	0.59 ± 0.02	1.57 ± 0.09	0.50 ± 0.03
	Hydrophobic–Polar	0.57 ± 0.04	1.67 ± 0.19	0.47 ± 0.04
	Polar–Polar	0.62 ± 0.05	1.21 ± 0.13	0.46 ± 0.05
	Polar–Hydrophobic	0.49 ± 0.04	1.56 ± 0.12	0.39 ± 0.03
	Small–Small	0.56 ± 0.03	1.54 ± 0.11	0.44 ± 0.03
	Small–Large	0.54 ± 0.04	1.37 ± 0.14	0.43 ± 0.04
	Large–Large	0.50 ± 0.05	1.62 ± 0.13	0.38 ± 0.04
	Large–Small	0.64 ± 0.02	1.46 ± 0.10	0.59 ± 0.03
	Aliphatic–Aliphatic	0.65 ± 0.02	1.25 ± 0.08	0.58 ± 0.03
INPS ^a^	Hydrophobic–Hydrophobic	0.61 ± 0.02	1.52 ± 0.08	0.49 ± 0.02
	Hydrophobic–Polar	0.57 ± 0.04	1.61 ± 0.15	0.40 ± 0.03
	Polar–Polar	0.40 ± 0.05	1.66 ± 0.15	0.15 ± 0.02
	Polar–Hydrophobic	0.42 ± 0.04	1.57 ± 0.11	0.23 ± 0.02
	Small–Small	0.57 ± 0.02	1.52 ± 0.09	0.39 ± 0.02
	Small–Large	0.49 ± 0.05	1.44 ± 0.15	0.32 ± 0.04
	Large–Large	0.39 ± 0.04	1.76 ± 0.14	0.20 ± 0.02
	Large–Small	0.59 ± 0.03	1.51 ± 0.10	0.41 ± 0.02
	Aliphatic–Aliphatic	0.67 ± 0.02	1.26 ± 0.08	0.55 ± 0.02
INPS-3D	Hydrophobic–Hydrophobic	0.65 ± 0.02	1.26 ± 0.07	0.47 ± 0.02
	Hydrophobic–Polar	0.64 ± 0.04	1.35 ± 0.13	0.43 ± 0.03
	Polar–Polar	0.55 ± 0.04	1.41 ± 0.15	0.21 ± 0.02
	Polar–Hydrophobic	0.55 ± 0.03	1.29 ± 0.10	0.28 ± 0.02
	Small–Small	0.62 ± 0.02	1.32 ± 0.08	0.39 ± 0.02
	Small–Large	0.56 ± 0.05	1.17 ± 0.12	0.33 ± 0.03
	Large–Large	0.52 ± 0.04	1.45 ± 0.14	0.25 ± 0.02
	Large–Small	0.67 ± 0.02	1.21 ± 0.09	0.45 ± 0.02
	Aliphatic–Aliphatic	0.71 ± 0.02	1.04 ± 0.07	0.52 ± 0.02
mCSM	Hydrophobic–Hydrophobic	0.62 ± 0.02	1.32 ± 0.07	0.36 ± 0.01
	Hydrophobic–Polar	0.69 ± 0.03	1.16 ± 0.09	0.43 ± 0.02
	Polar–Polar	0.67 ± 0.03	1.08 ± 0.11	0.38 ± 0.02
	Polar–Hydrophobic	0.67 ± 0.02	1.02 ± 0.07	0.37 ± 0.02
	Small–Small	0.62 ± 0.02	1.31 ± 0.08	0.33 ± 0.02
	Small–Large	0.61 ± 0.05	1.09 ± 0.12	0.31 ± 0.03
	Large–Large	0.65 ± 0.03	1.15 ± 0.10	0.37 ± 0.02
	Large–Small	0.73 ± 0.02	1.02 ± 0.06	0.49 ± 0.02
	Aliphatic–Aliphatic	0.69 ± 0.02	1.07 ± 0.07	0.44 ± 0.02
MAESTRO	Hydrophobic–Hydrophobic	0.63 ± 0.02	1.37 ± 0.08	0.51 ± 0.02
	Hydrophobic–Polar	0.64 ± 0.04	1.44 ± 0.14	0.59 ± 0.05
	Polar–Polar	0.58 ± 0.03	1.29 ± 0.14	0.38 ± 0.03
	Polar–Hydrophobic	0.63 ± 0.03	1.14 ± 0.08	0.41 ± 0.02
	Small–Small	0.64 ± 0.02	1.26 ± 0.08	0.48 ± 0.02
	Small–Large	0.61 ± 0.04	1.09 ± 0.09	0.45 ± 0.04
	Large–Large	0.52 ± 0.04	1.52 ± 0.13	0.36 ± 0.03
	Large–Small	0.69 ± 0.02	1.21 ± 0.07	0.59 ± 0.02
	Aliphatic–Aliphatic	0.71 ± 0.02	1.05 ± 0.05	0.62 ± 0.02
PoPMuSiC	Hydrophobic–Hydrophobic	0.61 ± 0.02	1.35 ± 0.08	0.40 ± 0.02
	Hydrophobic–Polar	0.58 ± 0.04	1.50 ± 0.15	0.38 ± 0.04
	Polar–Polar	0.46 ± 0.05	1.52 ± 0.18	0.24 ± 0.03
	Polar–Hydrophobic	0.58 ± 0.03	1.22 ± 0.09	0.36 ± 0.02
	Small–Small	0.61 ± 0.02	1.32 ± 0.08	0.39 ± 0.02
	Small–Large	0.53 ± 0.05	1.26 ± 0.13	0.29 ± 0.03
	Large–Large	0.46 ± 0.04	1.58 ± 0.14	0.25 ± 0.03
	Large–Small	0.66 ± 0.02	1.23 ± 0.09	0.48 ± 0.02
	Aliphatic–Aliphatic	0.68 ± 0.02	1.07 ± 0.06	0.47 ± 0.02
SDM	Hydrophobic–Hydrophobic	0.42 ± 0.02	2.44 ± 0.13	0.42 ± 0.03
	Hydrophobic–Polar	0.49 ± 0.04	2.05 ± 0.20	0.42 ± 0.05
	Polar–Polar	0.35 ± 0.04	1.81 ± 0.16	0.20 ± 0.03
	Polar–Hydrophobic	0.33 ± 0.03	2.62 ± 0.13	0.21 ± 0.02
	Small–Small	0.47 ± 0.02	2.44 ± 0.12	0.47 ± 0.03
	Small–Large	0.42 ± 0.04	1.77 ± 0.14	0.38 ± 0.04
	Large–Large	0.27 ± 0.04	2.12 ± 0.16	0.15 ± 0.02
	Large–Small	0.47 ± 0.02	2.61 ± 0.12	0.52 ± 0.03
	Aliphatic–Aliphatic	0.53 ± 0.03	2.17 ± 0.10	0.60 ± 0.03
DUET	Hydrophobic–Hydrophobic	0.62 ± 0.02	1.35 ± 0.07	0.46 ± 0.02
	Hydrophobic–Polar	0.70 ± 0.03	1.12 ± 0.10	0.49 ± 0.03
	Polar–Polar	0.67 ± 0.03	1.07 ± 0.10	0.41 ± 0.03
	Polar–Hydrophobic	0.68 ± 0.02	1.04 ± 0.07	0.40 ± 0.02
	Small–Small	0.64 ± 0.02	1.30 ± 0.08	0.43 ± 0.02
	Small–Large	0.63 ± 0.04	1.01 ± 0.11	0.39 ± 0.03
	Large–Large	0.64 ± 0.03	1.19 ± 0.10	0.38 ± 0.02
	Large–Small	0.72 ± 0.02	1.09 ± 0.06	0.58 ± 0.02
	Aliphatic–Aliphatic	0.68 ± 0.02	1.16 ± 0.07	0.58 ± 0.02

^a^ Sequence-based method.

**Table 6 ijms-24-12073-t006:** Sensitivity comparison of methods on the SKEMPI-SEQ-2388 and SKEMPI-3D-3775 datasets according to different types of amino acid mutations. The table provides the PCC, MSE, and slope of the fitting line, along with the standard deviation of each of them taken over 100 random selections of data points (see Section 4 for details).

Methods	Types of Amino Acid Mutation	SKEMPI-SEQ-2388		
		PCC	MSE	Slope
SAAMBE-SEQ	Hydrophobic–Hydrophobic	0.80 ± 0.03	0.94 ± 0.17	0.63 ± 0.04
	Polar–Polar	0.91 ± 0.02	0.99 ± 0.23	0.78 ± 0.05
	Polar–Hydrophobic	0.89 ± 0.01	0.49 ± 0.05	0.74 ± 0.03
	Small–Small	0.86 ± 0.02	0.46 ± 0.06	0.62 ± 0.03
	Small–Large	0.90 ± 0.02	1.13 ± 0.32	0.70 ± 0.06
	Large–Large	0.86 ± 0.03	1.59 ± 0.24	0.74 ± 0.04
	Large–Small	0.88 ± 0.02	0.69 ± 0.08	0.72 ± 0.03
	Aliphatic–Aliphatic	0.85 ± 0.04	0.92 ± 0.18	0.70 ± 0.05
	Aromatic–Aliphatic	0.90 ± 0.02	0.86 ± 0.21	0.66 ± 0.05
		**SKEMPI-3D-3775**		
		**PCC**	**MSE**	**Slope**
SAAMBE 3D	Hydrophobic–Hydrophobic	0.91 ± 0.01	0.49 ± 0.05	0.68 ± 0.02
	Polar–Polar	0.91 ± 0.01	0.98 ± 0.16	0.63 ± 0.03
	Polar–Hydrophobic	0.88 ± 0.01	0.6 ± 0.04	0.61 ± 0.01
	Small–Small	0.84 ± 0.01	0.46 ± 0.04	0.56 ± 0.02
	Small–Large	0.93 ± 0.01	0.83 ± 0.18	0.64 ± 0.03
	Large–Large	0.91 ± 0.01	1.01 ± 0.13	0.62 ± 0.02
	Large–Small	0.90 ± 0.01	0.60 ± 0.04	0.67 ± 0.01
	Aliphatic–Aliphatic	0.91 ± 0.01	0.60 ± 0.07	0.65 ± 0.02
	Aromatic–Aliphatic	0.91 ± 0.01	0.91 ± 0.12	0.57 ± 0.02
MutaBind2	Hydrophobic–Hydrophobic	0.88 ± 0.01	0.59 ± 0.06	0.67 ± 0.02
	Polar–Polar	0.92 ± 0.01	0.73 ± 0.07	0.76 ± 0.03
	Polar–Hydrophobic	0.89 ± 0.01	0.53 ± 0.04	0.65 ± 0.02
	Small–Small	0.83 ± 0.02	0.45 ± 0.04	0.59 ± 0.02
	Small–Large	0.92 ± 0.02	0.74 ± 0.09	0.71 ± 0.03
	Large–Large	0.91 ± 0.02	0.87 ± 0.11	0.74 ± 0.03
	Large–Small	0.90 ± 0.01	0.59 ± 0.05	0.68 ± 0.02
	Aliphatic–Aliphatic	0.91 ± 0.01	0.56 ± 0.07	0.70 ± 0.03
	Aromatic–Aliphatic	0.91 ± 0.02	0.68 ± 0.10	0.70 ± 0.03
mCSM-PPI2	Hydrophobic–Hydrophobic	0.92 ± 0.01	0.48 ± 0.07	0.67 ± 0.02
	Polar–Polar	0.92 ± 0.01	0.96 ± 0.14	0.66 ± 0.03
	Polar–Hydrophobic	0.92 ± 0.01	0.46 ± 0.04	0.66 ± 0.01
	Small–Small	0.86 ± 0.03	0.40 ± 0.07	0.61 ± 0.03
	Small–Large	0.89 ± 0.02	1.16 ± 0.21	0.56 ± 0.03
	Large–Large	0.88 ± 0.02	1.26 ± 0.20	0.60 ± 0.03
	Large–Small	0.95 ± 0.00	0.42 ± 0.03	0.70 ± 0.01
	Aliphatic–Aliphatic	0.93 ± 0.01	0.51 ± 0.09	0.68 ± 0.03
	Aromatic–Aliphatic	0.94 ± 0.01	0.66 ± 0.13	0.64 ± 0.03
BeAtMuSiC	Hydrophobic–Hydrophobic	0.46 ± 0.03	2.02 ± 0.18	0.29 ± 0.03
	Polar–Polar	0.35 ± 0.06	4.12 ± 0.61	0.15 ± 0.03
	Polar–Hydrophobic	0.31 ± 0.04	2.21 ± 0.17	0.16 ± 0.02
	Small–Small	0.31 ± 0.03	1.30 ± 0.01	0.16 ± 0.02
	Small–Large	0.13 ± 0.07	4.27 ± 0.73	0.04 ± 0.03
	Large–Large	0.19 ± 0.08	5.08 ± 0.66	0.09 ± 0.04
	Large–Small	0.47 ± 0.03	2.26 ± 0.19	0.28 ± 0.02
	Aliphatic–Aliphatic	0.28 ± 0.07	2.90 ± 0.46	0.15 ± 0.04
	Aromatic–Aliphatic	0.31 ± 0.07	3.42 ± 0.59	0.17 ± 0.05

**Table 7 ijms-24-12073-t007:** Sensitivity comparison of methods on S419 and ProNAB-237 datasets according to different types of amino acid mutations. The table provides the PCC, MSE, and slope of the fitting line, along with the standard deviation of each of them taken over 100 random selections of data points (see Section 4 for details).

Methods	Types of Amino Acid Mutation	S419		
		PCC	MSE	Slope
SAMPDI-3D	Polar–Hydrophobic	0.80 ± 0.02	0.51 ± 0.06	0.48 ± 0.02
	Large–Small	0.82 ± 0.03	0.48 ± 0.05	0.49 ± 0.02
mCSM-NA	Polar–Hydrophobic	0.39 ± 0.07	1.37 ± 0.18	0.33 ± 0.07
	Large–Small	0.33 ± 0.08	1.56 ± 0.22	0.3 ± 0.07
PremPDI	Polar–Hydrophobic	0.4 ± 0.05	1.75 ± 0.33	0.44 ± 0.06
	Large–Small	0.4 ± 0.06	1.48 ± 0.18	0.39 ± 0.06
		**ProNAB-237**		
		**PCC**	**MSE**	**Slope**
SAMPDI-3D	Polar–Hydrophobic	0.61 ± 0.05	1.23 ± 0.15	0.32 ± 0.03
	Large–Small	0.60 ± 0.06	1.45 ± 0.02	0.29 ± 0.04
mCSM-NA	Polar–Hydrophobic	0.40 ± 0.09	1.98 ± 0.36	0.33 ± 0.07
	Large–Small	0.47 ± 0.12	1.86 ± 0.40	0.33 ± 0.09
PremPDI	Polar–Hydrophobic	0.51 ± 0.05	1.70 ± 0.24	0.43 ± 0.06
	Large–Small	0.50 ± 0.07	1.84 ± 0.24	0.39 ± 0.07

## Data Availability

All databases are freely available for download from http://compbio/clemson.edu (accessed on 30 May 2023).

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
