# Peer review of "Predicting the Effect of Single Mutations on Protein Stability and Binding with Respect to Types of Mutations"

_ijms, 2023, doi:10.3390/ijms241512073_

Round 1

Reviewer 1 Report (Previous Reviewer 2)

I thank the authors for the manuscript corrections.

I have a comment about the conclusion of the manuscript? It seems that the conclusion should be supplemented with a comment regarding the occurrence of a significant number of mutations of any amino acid to alanine. Analyzing the data presented in Figures S1a, S1d, S1g, S1j, S1m, it is clear that the vast majority of the data were Xxx-Ala mutations. Such mutations can be classified both as SNV and non-SNV depending on what the original amino acid in the sequence was. Thus, a comment is needed from the authors showing the weakness of the paper (based on the available data), which inevitably focuses on the Xxx-Ala mutations.

A minor note: The acknowledgments are "Acknowledgements: We thank Clemson University Palmetto supercomputer". Is it too strange that authors thank the computer? You can thank the computer service employees for access to the computer, but I have not encountered a situation where you can thank the computer.

None

Author Response

We thank the reviewer for useful suggestions. 

Reviewr: I have a comment about the conclusion of the manuscript? It seems that the conclusion should be supplemented with a comment regarding the occurrence of a significant number of mutations of any amino acid to alanine. Analyzing the data presented in Figures S1a, S1d, S1g, S1j, S1m, it is clear that the vast majority of the data were Xxx-Ala mutations. Such mutations can be classified both as SNV and non-SNV depending on what the original amino acid in the sequence was. Thus, a comment is needed from the authors showing the weakness of the paper (based on the available data), which inevitably focuses on the Xxx-Ala mutations.

Ans: A sentence was added at the bottom of the conclusion section to reflect that.

Reviewer: A minor note: The acknowledgments are "Acknowledgements: We thank Clemson University Palmetto supercomputer". Is it too strange that authors thank the computer? You can thank the computer service employees for access to the computer, but I have not encountered a situation where you can thank the computer.

Ans: Corrected to “We thank Clemson University CCIT group for providing access to Clemson University Palmetto supercomputer.”

Reviewer 2 Report (Previous Reviewer 1)

The differences between the SNV, non-SNV and the whole dataset are not significant across the methods and datasets. There are two cases exhibiting relative big divergent (table 4, mCSM-NA on S419 and ProNAB-237), but they are equally bad on prediction for both SNV and non-SNV groups. Although it's informative to see the performs of different methods in predicting the folding and binding of proteins in various datasets, the mutation categorization did not provide more information. 

Author Response

We thank the reviewer for their comment.

Reviewer: 

The differences between the SNV, non-SNV and the whole dataset are not significant across the methods and datasets. There are two cases exhibiting relative big divergent (table 4, mCSM-NA on S419 and ProNAB-237), but they are equally bad on prediction for both SNV and non-SNV groups. Although it's informative to see the performs of different methods in predicting the folding and binding of proteins in various datasets, the mutation categorization did not provide more information.

Ans: While the difference between SNVs and non-SNSs are relatively small, there is clear trend that most methods perform worse on SNVs, and this is the main message of the paper.

This manuscript is a resubmission of an earlier submission. The following is a list of the peer review reports and author responses from that submission.

Round 1

Reviewer 1 Report

In this paper, the authors assessed various methods in the cases of SNVs and non-SNVs by comparing the PCC and MSE of predictions of folding or binding free energy changes vs experimentally measured values. I feel like the authors need more crafts on their manuscript and study design. 

1.     The mutation classification (SNV vs non-SNV) is over simplified. Polar to non-polar and polar to polar mutations may have very different effect. As the authors mentioned, the results are influenced by the chemical nature and size of the mutations, like “SNV cases in S2648 are dominated by hydrophobic to hydrophobic and small to small mutations while non-SNVs by large to small and polar to hydrophobic mutations whereas SNV cases in both SKEMPI-SEQ-2388 and SKEMPI-3D-3775 is dominated by small-small and polar to hydrophobic mutations while non-SNV mutations from large to small amino acids.”

2.     The numbers of mutations in different categories are not the same. A more rigorous way for the comparison may be required to statistically assess the significance of the difference.   

3.     Although the authors used different predictors, they claimed they did not intend to compare the performance. So there is no extensive benchmarking in this manuscript. Well, the title “assessment of leading algorithms performance” may make readers expect more on the benchmarking front.   

In terms of wording, the authors tend to use vague/non-scientific expressions. Just to name some examples: 

1.      “methods with adjustable parameters or machine learning algorithms” is not a proper way to name the categories; 

2.     “Pearson correlation …. are less impressive than for non-SNVs”, what does “less impressive” mean? and what values were the PCCs calculated from? 

3.     The authors wanted to explore how the different type of mutations affect protein stability and binding. Folding and binding are two different processes. It’s confusing in the Figure1 the authors plotted the distribution of binding free energy changes but stated in the legend that the plots were for both folding and binding. 

Reviewer 2 Report

In my opinion, the manuscript submitted for review is not suitable for publication in its present form.

Main objections to work:

1. The authors conclude that the algorithms used to predict the thermodynamic stability of mutant proteins perform much better in the case of non-SNV mutations. I understand that the authors observe such a relationship, but is it an overarching rule, or is it due to other reasons, such as the number of available data in individual groups. In order to really draw constructive conclusions, the authors should conduct an additional analysis in which they will randomly divide the available data set into parts (two or more) and for each part they will perform an analysis and compare the results obtained in this way with those presented in the paper. The differences between the non-SNV and SNV groups are very small and it is not known whether the differences are statistically significant, additional analysis with randomly selected datasets should provide an answer. The analyzes and conclusions presented in the work are not based on reliable data.

2. There are no clearly described datasets. The authors describe the entire data sets, provide information on the total number of mutations analyzed, but do not specify how it is broken down into non-SNV and SNV groups? This can only be seen by laboriously adding up the data provided in Supplemental Data (e.g. Fig S1e, Fig S1h). This is extremely important information (the number of data in each group) needed to statistically evaluate the results (see note 1).

3. The authors devote a significant part of the text and table figures to analyzing the frequency of mutations, which amino acids are replaced with others, etc. (eg Fig. S1). The analysis of this data boils down to the perfunctory statements contained in the Dissusion section). Presenting these data would make sense if the authors had analyzed the algorithms in terms of their effectiveness, e.g. in the case of a mutation of a hydrophobic residue to another hydrophobic residue, or other types of mutations. Such an analysis (the effectiveness of algorithms in terms of the type of mutation) would be extremely valuable and probably no one has conducted such an analysis.

4. Only from certain premises one can guess that the authors analyze only mutations for proteins from the species Homo Sapiens. If the work was intended to test the general efficiency of algorithms, then why did the authors limit themselves to one genre. In addition, the authors rely on quite old datasets. For example, the dataset described as S2648 (reference 55) based on the ProTherm database and dates from 2009. The ProTherm database has been significantly expanded and the 2021 edition (ref 15) contains over 2 times more data (single mutations) than the 2006 edition.

5. There is no clear justification in the paper why only single mutations were analyzed? The title of the paper does not specify that the analysis concerns only single point mutations.

6. Descriptions of algorithms and databases are included in the methods and occupy a significant part of the introduction. On the other hand, the introduction does not clearly introduce the merits of the conducted research.

None